# Implications of Model Indeterminacy for Explanations of Automated Decisions

**Marc-Etienne Brunet**
University of Toronto
Vector Institute

mebrunet@cs.toronto.edu

**Ashton Anderson**
University of Toronto
Vector Institute

ashton@cs.toronto.edu

**Richard Zemel**
University of Toronto
Columbia University
Vector Institute

zemel@cs.toronto.edu

## Abstract

There has been a significant research effort focused on explaining predictive models, for example through post-hoc explainability and recourse methods. Most of the proposed techniques operate upon a single, fixed, predictive model. However, it is well-known that given a dataset and a predictive task, there may be a multiplicity of models that solve the problem (nearly) equally well. In this work, we investigate the implications of this kind of model indeterminacy on the post-hoc explanations of predictive models. We show how it can lead to *explanatory multiplicity*, and we explore the underlying drivers. We show how predictive multiplicity, and the related concept of epistemic uncertainty, are not reliable indicators of explanatory multiplicity. We further illustrate how a set of models showing very similar aggregate performance on a test dataset may show large variations in their local explanations, i.e., for a specific input. We explore these effects for Shapley value based explanations on three risk assessment datasets. Our results indicate that model indeterminacy may have a substantial impact on explanations in practice, leading to inconsistent and even contradicting explanations.

## 1 Introduction

Data-driven decision making has become a preferred practice within organizations. When the most important outcomes are readily quantifiable, entire decision making processes may even be automated using predictive models fit to historic data. With so much to be gained from accurate predictions, powerful, modern, "black-box" machine learning (ML) models have obvious appeal. Clearly, however, it is understood that deciding whether to deny access to products like insurance and credit should not be taken lightly. Governing regulation may require a justification to be provided in conjunction with a decision, such as one from a set of adverse action codes [34]. Even where such regulation is not required, there is increasing pressure to make automated decisions more transparent [31]. As a result, considerable interest has recently been devoted to explaining ML model decisions. Academics have proposed numerous new techniques [2], new businesses have been founded to address this need, and regulators have invested significant effort to provide guidance [21].

Thus far, the focus of most machine learning explanation and transparency research has been on a single learned predictive model. However, it is well-known that given a dataset and a predictive task, there may be a multiplicity of models that solve the problem (nearly) equally well[5; 6; 30]. What are the implications of this kind of model indeterminacy for the explanations of automated decisions? It has been demonstrated that a set of models with very similar performance on a test set may still show significant *predictive multiplicity* [20], which is to say, the models make conflicting predictions on individual data points. Quantifying this kind of model uncertainty in machine learning predictions is the subject of considerable research interest. For instance it is central to work in Bayesian modelling, and to discussions of epistemic uncertainty [16]. But to what extent are *explanations* affected?

36th Conference on Neural Information Processing Systems (NeurIPS 2022).

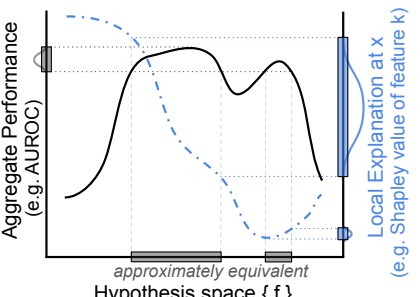 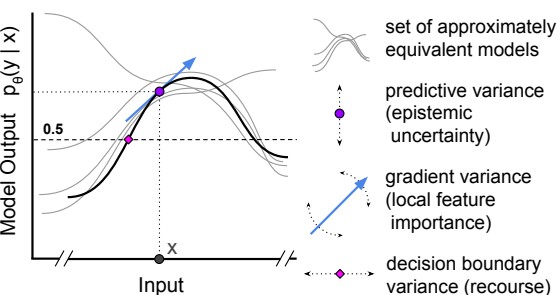

Figure 1: [Left] An aggregate performance metric like AUROC (black solid line) and a local explanation like a Shapley value (blue dot-dash line) depend on a model's predictive function, $f(\boldsymbol{x})$, in different ways. Even if we see low variance in the aggregate performance across a set of models, we may see high variance in the individual local explanations. [Right] Visualizing how model indeterminacy may affect predictive multiplicity (epistemic uncertainty), local feature-based explanations, and recourse separately. Consider a set of approximately equivalent models, and a point in the input space, $\boldsymbol{x}$. The differences we may see in the model output probability at $\boldsymbol{x}$ are distinct from the differences we may see in their gradients (with respect to $\boldsymbol{x}$). The location of the nearest decision boundary may also change independently.

In practical applications, where the stakes are high, might model indeterminacy jeopardize the trustworthiness of an automated decision system by undermining transparency efforts? We explore these questions in the context of risk assessment, i.e., insurance and credit applications, focusing on a well-established class of explanation techniques which use Shapley values [7; 19; 22], but our analysis applies to a broader set of systems and methods.

**Contributions**    In this work, we introduce the notion of *explanatory multiplicity*, and study it both analytically and experimentally. We show how a set of models with very similar test-set performance may have large variations in their local explanations (Figure 1 left). We further show how predictive multiplicity, is not required for—or even indicative of—explanatory multiplicity (Figure 1 right). This is because in general, explanation methods are affected not just by the value of the predictive function at a point, but by its shape. We connect this analysis to an existing measure of model uncertainty: epistemic uncertainty [16]. Through experimentation, we explore explanatory multiplicity on three risk assessment datasets. We measure the changes in the decisions and corresponding explanations that users would receive if an automated decision system switched between approximately equivalent models. Our results indicate that model indeterminacy may have a substantial impact on explanations in practice, leading to inconsistent and even contradicting explanations.

## 2   Setup and related work

**Problem setup**    We consider the situation whereby a user (applicant) is processed by a (partially) automated decision-making system, which ultimately accepts or rejects them. We assume the system is built around a machine learning model that has been trained on relevant historical data. The model makes a prediction which is used in the decision to accept or reject the applicant, possibly in conjunction with other eligibility criteria or human expertise. However, we are concerned with explaining the *model prediction* to the user. The explanation should be *local*, in the sense that it should be specific to the user's input data, rather than describing a *global* property of the model.

Assume we have a dataset of user-data outcome pairs, $D = \{\boldsymbol{x}_i, y_i\}_{i=1}^N$, with multi-dimensional input, $\boldsymbol{x} = (x_1, x_2, ..., x_M) \in \mathcal{X}$, and label $y \in \mathcal{Y}$. We denote the set of features (dimensions) of $\mathcal{X}$ as $\mathcal{M} = \{1, 2, ..., M\}$. Let $f(\boldsymbol{x})$ be a predictive function from a set of candidate functions (hypothesis space), $\mathcal{F}$, mapping the users' data, $\mathcal{X}$, to an output prediction score $z \in \mathcal{Z} \subseteq \mathbb{R}$ used in determining their acceptance. A typical scenario may be that $\mathcal{Y} = \{0, 1\}$, with $\mathcal{F}$ modelling the probability $p(y = 1|\boldsymbol{x})$, in which case $\mathcal{Z} = [0, 1]$. We suppose that the system produces *local*, vector-valued explanations, of the form $\boldsymbol{\phi} : (\mathcal{X}; \mathcal{F}) \mapsto \mathbb{R}^M$. This may be a *feature-importance* vector such as the gradient [3], $\nabla_{\boldsymbol{x}} f$, or those generated by LIME [27] or SHAP [19]. Alternatively, this may be an *action* vector determined by an algorithmic recourse method [32], in which case we

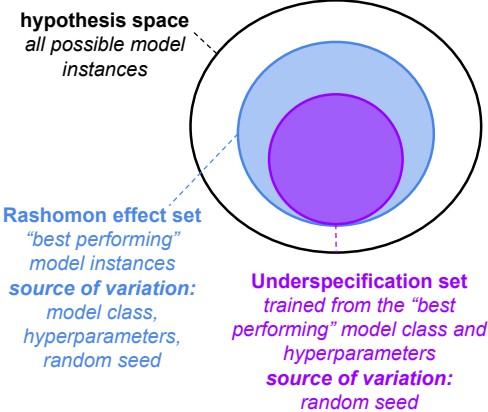

**hypothesis space**
*all possible model instances*

**Rashomon effect set**
*"best performing" model instances*
***source of variation:***
*model class, hyperparameters, random seed*

**Underspecification set**
*trained from the "best performing" model class and hyperparameters*
***source of variation:***
*random seed*

Figure 2: Model indeterminacy. The model instance that ends up in production is often not fully determined by the data, modelling objectives, and constraints. There is likely to be a set of different models showing nearly indistinguishable validation performance; this is known as the Rashomon effect. Even if the model class, hyperparameters, and optimization method are all specified, randomness in training may lead to substantial functional differences in the final model instance; this is known as underspecification. Despite this potential for variability, most existing explanation techniques operate on a single model instance.

further require $(\boldsymbol{\phi}(\boldsymbol{x}; f) + \boldsymbol{x}) \in \mathcal{X}$. In either case, we let $\phi_k(\boldsymbol{x}; f)$ denote the component of the explanation relevant to input feature $k \in \mathcal{M}$. We assume there exists a dataset dependent *model selection criteria*, $S_D : \mathcal{F} \mapsto \{0, 1\}$, which filters $\mathcal{F}$ down to a set of *approximately equivalent models*, $\mathcal{R} = \{f \in \mathcal{F} : S_D(f) = 1\}$. For example, the selection criteria may consist of limiting $\mathcal{R}$ to only include models which score above a certain threshold on an aggregate performance metric.

We are interested in the set of possible explanations that could be given to a user across all approximately equivalent models, $\{\boldsymbol{\phi}(\boldsymbol{x}; f) : f \in \mathcal{R}\}$. We are particularly concerned with situations whereby there exist $f, f' \in \mathcal{R}$ such that $\phi_k(\boldsymbol{x}; f) < 0 < \phi_k(\boldsymbol{x}; f')$, as this would constitute *contradicting* explanations. A user processed by $f$ would be explained that feature $k$ contributed positively to their assessment (or should be increased in the case of recourse), but had the same user been processed by $f'$, they would have been explained that feature $k$ contributed negatively to their assessment (or should be decreased).

### 2.1 Model indeterminacy: Underspecification and the Rashomon effect

Model indeterminacy arises when there are many *approximately equivalent* predictive functions within the hypothesis space. We say that two models are approximately equivalent if, in a given context, we cannot justify why we should use one instead of the other—both would satisfy our model selection criteria. We consider two types of model indeterminacy: *underspecification*, as studied by D'Amour et al. [6], and the *Rashomon effect*, as originally described by Breiman [5]. They arise in two slightly different commonplace contexts, but each may lead to sets of approximately equivalent models. The relationship between them is depicted in Figure 2. Underspecification arises when the model specification and training set are not enough to fully determine the model parameters. The training set, model class, hyperparameters, and optimization method are fixed, but the intentional use of randomness during training can lead to sets of approximately equivalent model parameters that effectively differ only because of the choice of random seeds. When the Rashomon effect is observed during model development, many models (possibly of different classes) show approximately equal performance. Choosing which model among them to use in deployment becomes an arbitrary choice; thus this set of best models may be considered approximately equivalent. The Rashomon effect is more general than underspecification, as it does not require the model specification to be held constant. These types of model indeterminacy are very common in practical settings [6; 5].

### 2.2 Shapley value based explanations

We focus on the class of *local, feature-based* explanations that use Shapley values to attribute importance to the model's input features. It has been shown that explanations satisfy a set of desirable criteria if and only if they are computed using Shapley values [19]. This has put them in the spotlight for regulatory purposes [34; 21]. The core idea of using Shapley values for explanations is to explore all the possible subsets of features that the model could use to make a prediction, $\mathcal{S} \subset \mathcal{M}$, where $\mathcal{M}$ is the set of all input features, and to calculate the average marginal effect that including feature $k$ into $\mathcal{S}$ has on that prediction. There are several related Shapley-based methods, which vary in how they include/exclude features from a prediction. We present a unifying formulation introduced by

Merrick and Taly [22]. Let $\boldsymbol{u}(\boldsymbol{x}, \boldsymbol{r}, \mathcal{S})$ be a composite input function, which takes the values of $\boldsymbol{x}$ for all the features in $\mathcal{S} \subset \mathcal{M}$, and otherwise takes the values of $\boldsymbol{r}$, some other reference input; recall $|\mathcal{M}| = M$. The Shapley value for feature $k$ given input $\boldsymbol{x}$ can then be written as

$$\phi_k(\boldsymbol{x}; f) = \frac{1}{M} \sum_{\mathcal{S} \subseteq \mathcal{M} \setminus \{k\}} \binom{M-1}{|\mathcal{S}|}^{-1} \mathbb{E}_{R \sim \mathcal{D}_{\text{ref}}} \left[ f\left(\boldsymbol{u}\left(\boldsymbol{x}, R, \mathcal{S} \cup \{k\}\right)\right) - f\left(\boldsymbol{u}\left(\boldsymbol{x}, R, \mathcal{S}\right)\right) \right]. \quad (1)$$

The sum is over all subsets of features (excluding feature $k$). Each term is the marginal change in the prediction when including feature $k$, divided by the binomial coefficient $M-1$ choose $|S|$. The reference distribution, $\mathcal{D}_{\text{ref}}$, is chosen to zero out the contribution of the excluded features. Typically, some variant of the input distribution is used. By simply varying the choice of reference distribution, we can recover several different Shapley-based explanation methods [22], including those used in popular techniques like KernelSHAP [19] and Quantitative Input Influence [7].

## 2.3 Uncertainty quantification

Model indeterminacy can be examined through the lens of uncertainty quantification. A popular, Bayesian-inspired framework used in this pursuit subdivides uncertainty into two categories: aleatoric and epistemic [14]. Epistemic uncertainty aims to quantify the uncertainty in the model of the data generating process, and thus can be reduced by gathering more data. Aleatoric uncertainty aims to quantify the inherent (irreducible) noise in the data generating process. Kirsch et al. [16] propose that the uncertainty in the prediction at input $\boldsymbol{x}$ of a Bayesian model (with parameters $\boldsymbol{\theta}$) can be decomposed and categorized as

$$\underbrace{\mathbb{I}[Y; \boldsymbol{\theta}|\boldsymbol{x}, \mathcal{D}]}_{\text{epistemic}} = \underbrace{\mathbb{H}[Y|\boldsymbol{x}, \mathcal{D}]}_{\text{predictive}} - \underbrace{\mathbb{E}_{p(\boldsymbol{\theta}|\mathcal{D})}[\mathbb{H}[Y|\boldsymbol{x}, \boldsymbol{\theta}]]}_{\text{aleatoric}} \quad (2)$$

$$\approx \mathbb{H}\left[\frac{1}{n} \sum_{i=1}^{n} f(\boldsymbol{x}; \boldsymbol{\theta}_i)\right] - \frac{1}{n} \sum_{i=1}^{n} \mathbb{H}[f(\boldsymbol{x}; \boldsymbol{\theta}_i)] \quad (3)$$

where $\mathbb{H}[Y|x, \mathcal{D}]$ is the predictive entropy given the training data $\mathcal{D}$. The mutual information term, $\mathbb{I}[Y; \boldsymbol{\theta}|x, \mathcal{D}]$, which corresponds to the epistemic uncertainty, can be obtained by subtracting the aleatoric uncertainty from the predictive entropy. When we have access to samples from $p(\boldsymbol{\theta}|\mathcal{D})$, $\{\boldsymbol{\theta}_i\}_{i=1}^{n}$, and $f$ models the probability $p(Y = y|\boldsymbol{x})$, we can estimate Equation 2 with Equation 3 [24]. The epistemic uncertainty at $\boldsymbol{x}$ is high when the individual model instances, $\{f(\boldsymbol{x}; \boldsymbol{\theta}_i)\}_{i=1}^{n}$, confidently disagree at $\boldsymbol{x}$. We use epistemic uncertainty as a principled way of measuring the variation in the prediction probabilities in sets of approximately equivalent models.

## 2.4 Related work

Local explanation methods have already attracted some criticism with regards to their robustness. For example, Alvarez-Melis and Jaakkola [1] have noted a lack of robustness to small perturbations of the input being explained, while Lakkaraju and Bastani [18] show how this can be exploited to generate manipulative explanations. The effect of model indeterminacy on explanations has been explored in a philosophical sense by Hancox-Li [11]. Rudin [28] has argued against relying on post-hoc explanations of black box models in high stakes decision-making altogether. There has also been research into how the Rashomon effect can impact model interpretability. Semenova et al. [30] quantify the Rashomon effect, and differentiate it from existing ideas in the study of model complexity. Fisher et al. [9] consider the Rashomon set of "all well-performing models" within a specified class, and explore the extent to these models are necessarily dependent on any particular feature. They consider a *global* measure of feature importance called model reliance (MR), examining its range over the Rashomon set for each of the individual input features. Dong and Rudin [8] build on this work by characterizing the MR relationship *between* different features within the Rashomon set, using what they call the variable importance cloud (VIC). This set of work is highly relevant to our treatment of explanatory multiplicity, but it does not directly address the popular *local* explanation methods which explain the prediction for a *specific user*, i.e., a specific $\boldsymbol{x}$. These are currently the main candidates for explanations in practice. In contrast, global measures of feature importance like MR are tied to how important a feature is for a model's aggregate performance on a whole dataset. Recently, Black et al. [4] examine the inconsistencies in model predictions and simple gradient explanations that arise from underspecification and leave-one-out changes to the training data. They propose using selective ensembles to make both predictions and explanations more consistent.

# 3 Explanatory multiplicity

We now examine how model indeterminacy can bring about explanatory multiplicity. First consider the relative consistency of a performance metric, e.g., accuracy or AUROC, versus the value of a local explanation across a set of approximately equivalent models, $\{\phi(\boldsymbol{x}; f) : f \in \mathcal{S}\}$. Regardless of the exact methods employed, the local explanation $\phi(\boldsymbol{x}; f)$ will almost certainly depend on $f$ differently than the aggregate performance metric. Therefore, even if a set of approximately equivalent models show very similar aggregate performance, we should not expect consistency in their individual local explanations. The difference in dependency on $f$ might transform a tight distribution over the performance metric into a wide, possibly multi-modal, distribution over the Shapley value for a feature, $\phi_k(\boldsymbol{x}; f)$. This is illustrated in Figure 1 (left).

Now consider how the form of the predictive function may vary in the region around some input $\boldsymbol{x}$ across the approximately equivalent models. This is illustrated in Figure 1 (right). We focus on how three local characteristics of the predictive function may vary within the set of approximately equivalent models. First, there can be variance in the value of the predictive function $f(\boldsymbol{x})$. Therefore, we may see considerable disagreement at any individual point in the input space. Strong disagreement would drive high epistemic uncertainty, per Equation 2. Second, there can be variance in the gradient, $\nabla_{\boldsymbol{x}} f(\boldsymbol{x})$, (or the discrete equivalent to this quantity if $f$ is not differentiable). The gradient is an indicator of local feature importance. It is used directly in simple explanation techniques [3], and it is closely connected to several others [29], including Shapley-value based methods in linear models. Finally, there can be variance in the position of the nearest decision boundary. This has implications for counterfactual explanations [33] and algorithmic recourse [32]. It is possible that the nearest decision boundary to a user shift significantly between approximately equivalent models. This could nullify the validity of the recourse provided. Importantly, these three characteristics may vary independently. Therefore, while epistemic uncertainty may be a useful construct for indicating when we should expect to see variance in predictions, it is not indicative of when explanations will vary. Our experimental results presented below support this.

**Illustrative example**     To illustrate these effects, we train 200 neural networks (MLPs) on a synthetic dataset ($\boldsymbol{x} \in \mathbb{R}^2$, $y \in \{0, 1\}$). These model instances differ only because of differences in the random seeds used in training. We limit our consideration to the 3% highest-performing model instances, selecting those ranging between 93.3% and 93.6% validation accuracy. This constitutes a set of approximately equivalent models. Because the models differ only in the random seeds, underspecification is the source of model indeterminacy here. We plot the models' output (softmax probability) over a region of the input space in Figure 15 in Appendix E. Their functional forms may change considerably from one model instance to the next, in some cases, causing the nearest counterfactual boundary to shift significantly. A single model instance along with the (iid) test set is shown in Figure 3 (left). The predictive multiplicity at each point in that region of the input space is calculated using the epistemic uncertainty in the set of approximately equivalent models (Equation 2 explained below) and is shown in Figure 3 (center). Notice it increases as we move toward regions of the input space for which we have less data. Finally, we compute the mean pairwise cosine similarity of the gradients (with respect to $\boldsymbol{x}$) between models over the same region of the input space, shown in Figure 3 (right). We see that predictive multiplicity is a poor indicator of the similarity of the gradients across approximately equivalent models. For instance, consider the point at the orange star. We have low epistemic uncertainty (corresponding to inter-model agreement in predictions), yet the gradients across the models at this point are very dissimilar (orange arrows).

**Analysis in a linear setting**     In a linear model class, it is possible to determine the precise conditions under which model indeterminacy can lead to contradicting explanations. We present a brief summary of this analysis here, and refer the reader to Appendix C for more details. Consider the case where $\boldsymbol{x} \in \mathbb{R}^M, y \in \mathbb{R}$, and $\mathcal{F}$ is the set of ridge regression models of the form $f(\boldsymbol{x}; \boldsymbol{\theta}) = \boldsymbol{\theta}^T \boldsymbol{x}$, with square error loss and $\ell_2$ penalty $\alpha > 0$. The Rashomon set of all $\boldsymbol{\theta}$ which parameterize models within $\varepsilon$ of the optimal loss, is an ellipsoidal region, $\Theta_\varepsilon$, centered at the OLS solution, $\boldsymbol{\theta}^*$. We are interested in circumstances whereby the sign of a Shapley value reverses within this Rashomon set, i.e., $\exists \, \boldsymbol{\theta}, \boldsymbol{\theta}' \in \Theta_\varepsilon$ s.t. $\phi_k(\boldsymbol{x}; \boldsymbol{\theta}) > 0 > \phi_k(\boldsymbol{x}; \boldsymbol{\theta}')$.

It can be shown that the Shapley value of feature $k$ in this ridge regression setting is $\phi_k(\boldsymbol{x}; \boldsymbol{\theta}) = \theta_k x_k$. Since $\boldsymbol{x}$ is fixed for a user, $\phi_k$ changes sign if and only if $\theta_k$ does. In the simple case where the features of $\boldsymbol{x}$ are independent, the ellipsoidal region $\Theta_\varepsilon$ is axis-aligned. We show that $\theta_k$ is bounded in an interval centered at $\theta_k^*$, having width $2(\varepsilon/(\frac{1}{N} \sum_n x_{k,n}^2 + \alpha))^{1/2}$. For a fixed $\varepsilon$ relaxation from

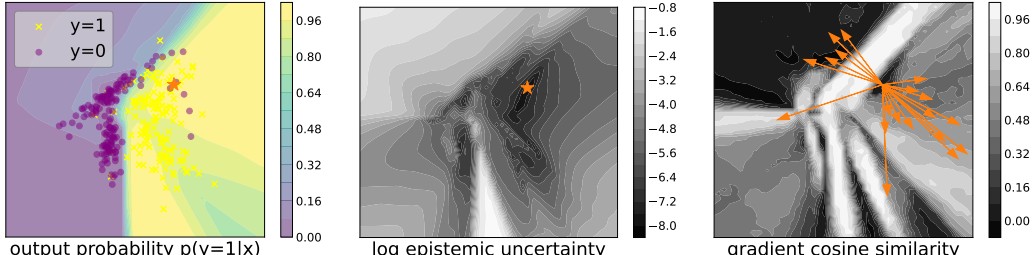

Figure 3: [Left]: Softmax probability of an MLP over a region of the input space. The test set is depicted as purple circles and yellow Xs. [Center]: Epistemic uncertainty over the same region of the input space is computed using a set of approximately equivalent model instances arising from the same training pipeline. [Right]: Mean pairwise cosine similarity of the gradients (w.r.t. the input) between models over the same region of the input space. The orange arrows are the gradients for (a subset of) the individual models at the point indicated by the orange star (left and center).

optimality and regularization coefficient $\alpha$, the width of this interval decreases with the variance of the feature $k$. So $\phi_k$ is a candidate for sign reversal if $x_k$ is a low variance independent feature, and $\theta_k^*$ is near zero.

Consider now if features $j$ and $k$ are correlated, with $\text{Cov}(x_j, x_k) > 0$, but all other features are independent. The ellipsoidal region is aligned for all axes $i \notin \{j, k\}$, but in the $j, k$ plane, the cross-sectional ellipse is rotated. In this case, we show that $\theta_k$ is also bounded in an interval centered at $\theta_k^*$. If we fix $\varepsilon$, $\alpha$, and the variance of features $j$ and $k$, we show that the width of this interval increases with $\text{Cov}(x_j, x_k)$. Intuitively, this tells us that problems for which the input features are strongly correlated are more prone to contradicting explanatory multiplicity. However, in complex hypothesis spaces, it is considerably more difficult to analyse explanatory multiplicity theoretically. We instead explore it through experimentation.

## 4 Experiments

To explore the extent to which model indeterminacy may impact the consistency of explanations in a practical setting, we conduct a series of experiments. We use three different (binary) risk assessment datasets (all available on Kaggle): UCI Credit Card [35], Give Me Some Credit, and Porto Seguro's Safe Driver Prediction. Their details can be found in Appendix B.1. In each case, we let $y = 1$ indicate the "bad" outcome: missed payment, serious delinquency, and insurance claim filed, respectively. We fit predictive models to each of these datasets, and consider their use in an automated decision system that filters out applicants (users) who pose an unacceptably high risk of a bad outcome. We then measure how the predictions and corresponding explanations are affected by the Rashomon effect and underspecification. Our experiments were conducted on a GPU accelerated computing cluster. ML models were written in PyTorch [26], and the analysis used NumPy [12] and Matplotlib [13] [1] [2].

**Preprocessing and model selection** We first split each dataset into a development and a holdout set (70 / 30), and apply one-hot encoding and standard scaling. We then run a model selection process with three model classes: logistic regression (LR), multi-layer perceptron (MLP), and a tabular ResNet (TRN) recently proposed by Gorishniy et al. [10]. We sweep through a range of hyperparameter settings, trying a total of 408 model-hyperparameter configurations per dataset. For each configuration, we pick a random seed and use it to control a shuffled split of the development dataset into train and validation sets (70 / 30). This seed also controls the randomness used in training (optimization). We fit the models using Adam [15] with a patience-based stopping criteria on the validation set. We also up-weight the rare class, creating a balanced loss. We repeat this process with 3 random seeds per configuration, obtaining a total of 1224 model instances per dataset.

---

[1]These packages are publicly available under BSD-style licenses.

[2]Experimental source code will be made available at `github.com/mebrunet/model-indeterminacy`

**Forming sets of approximately equivalent models**     To explore the effect of underspecification, we identify the best model-hyperparameter configuration and generate many approximately equivalent model instances from it. We refer to these as *underspecification sets*. Specifically, we use the AUROC on the validation sets (averaged over 3 random seeds) to determined the best model-hyperparameter configuration for each dataset. We then train 100 models instances using that configuration. In this final training, we fix the number of epochs, and use the full development dataset for each. Thus the *only* source of variation between model instances is the random seed controlling initialization and mini-batch order. To explore the impact of the Rashomon effect, we simply skim off the top 3% of model instances (36 of 1224) from the model selection sweep, ranked according to their AUROC performance on their validation sets. We refer to these as *Rashomon effect sets* as they stray slightly from the definition of a "Rashomon set" by Semenova et al. [30][3]. These sets include model instances from different model-hyperparameter configurations, and different controlling random seeds. After forming a set of either type, we use the holdout data to ensure that none of the model instances show unusually low performance out-of-sample.

**Computing explanations**     For each dataset, we consider a subset of 1000 instances from the holdout set. We generate Shapley value explanations on these points for each of the approximately equivalent models. Notice that in Equation 1 there is an expectation over the reference distribution, $R \sim \mathcal{D}$, as well as a sum over all subsets of features in $\mathcal{M}$ which grows as $\mathcal{O}(2^{|\mathcal{M}|})$. In practice this sum must be approximated for inputs with even a modest number of features, and the expectation is converted to an empirical mean. Fortunately, there has been considerable effort put into developing estimators for Shapley values. In this work we use a simple antithetic sampling approach described in [23]. Importantly, when comparing Shapley-based explanations across models we are careful to use precisely the same set of samples, thereby ensuring that if two models agreed pointwise everywhere in their domains, their explanations would be identical [4]. We consider two different reference distributions, $\mathcal{D}$ in Equation 1, each corresponding to a different type of Shapley explanation. The first is a point mass at the mean input, $\mathbb{E}[\boldsymbol{x}]$. In the language of Merrick and Taly [22], this amounts to considering a "single-reference game"; the explanations contrast the user with the "average applicant". We refer to this as $\mathcal{D}_{\mathrm{mean}}$. We also consider the input distribution; this amounts to explaining with KernelSHAP. As is typical, to limit compute time, we approximate the input distribution using a small subset of the training data. We randomly select 100 reference inputs, and we reuse them for each explanation to ensure this is not a source of variation. We refer to this as $\mathcal{D}_{\mathrm{input}}$.

## 4.1   Quantifying Consistency in Explanations

To quantify the consistency of explanations we focus on the top-k (in magnitude) Shapley values explaining a user's predicted risk (of a bad outcome). These correspond to the features which are most significant to the prediction. It is recommended to limit feature-based explanations to a subset of the most important features [27], thus these are most likely what would be shown to the user. If they are positive-valued, they indicate the feature adds to the applicant's predicted risk. If they are negative-valued, they indicate the feature reduces that risk. We consider three metrics to quantify the consistency of the explanations. Each of these metrics takes in a pair of explanations and outputs a numerical value. For each user, we average these metrics over the explanations from every pair of approximately equivalent models ($|\mathcal{R}|$ choose 2 pairs).

**Signed Set Disagreement (SSD)**     We first consider a binary metric which takes value $0$ if the sets of top-k features agree between two explanations, and a value of $1$ if they do not, i.e., they disagree. For agreement within the sets of top-k, we require that the sign of the Shapley values be the same, but not their numerical values or relative order. For example, suppose explanation A has Shapley values $\{\phi_1 : 0.4, \phi_2 : 0.7, \phi_3 : -0.9, \phi_4 : -0.1\}$, but explanation B has $\{\phi_1 : -0.5, \phi_2 : 0.8, \phi_3 : -0.7, \phi_4 : 0.3\}$. Their top-1 Shapley values disagree: $\{\phi_3\}$ vs. $\{\phi_2\}$, their top-2 Shapley values agree: $\{\phi_2, \phi_3\}$, but their top-3 Shapley values disagree because $\phi_1$ does not have the same sign in both explanations. When averaged over pairs of approximately equivalent models, the SSD estimates the probability that a user would see a disagreement in the set of (signed) top-k explanatory features if the predictive model were switched to another approximately equivalent model.

**Contradicting Direction of Contribution (CDC)**     We are most concerned about contradicting explanations, as these can be confusing or even harmful for a user. For this we consider another

---

[3]We elaborate on this difference in Appendix B.3

[4]Further information about this estimate is available in Appendix F.

binary metric which takes value 1 if at least one of the top-k Shapley values changes sign (direction of contribution) across two explanations, and takes value 0 otherwise. Thus the metric takes value 1 if there is at least one top-k feature, say feature-j, which is *adding to* the user's predicted risk in explanation A, but *reducing* their risk in explanation B. (Or vice-versa, going from reducing to adding). We require feature-j to be in the top-k features of explanation A or B, not necessarily both. When averaged over pairs of approximately equivalent models, the CDC estimates the probability that a user would see a top-k explanatory feature change direction of contribution if the predictive model were switched to another approximately equivalent model.

**Sign Agreement (SA)**     We also make use of the Sign Agreement (SA) metric proposed by Krishna et al. [17]. This metric computes the fraction of top-k features that are not only common between two explanations, but also share the same sign (direction of contribution) in both the explanations. It takes fractional values from $\{0, \frac{1}{k}, \frac{2}{k}, ..., 1\}$. Note that SSD indicates if SA $< 1$.

# 5   Results

Here we present the results of our experiments. Note that our figures focus on $\mathcal{D}_{\text{input}}$, underspecification, and the UCI Credit Card dataset. The corresponding figures for $\mathcal{D}_{\text{mean}}$, the Rashomon effect, and other datasets are presented in Appendix D. We tabulate the results of the hyperparameter sweep and set formation in Table 2, in Appendix B.2. The probability that two approximately equivalent models disagree on a user acceptance decision is a function of the decision threshold (risk appetite). With decision thresholds set to a 50% user acceptance rate, disagreement ranges between 4.35–12.7% across our experimental setups. We report more details on the consistency of decisions across a range of user acceptance rates in Appendix D.1.

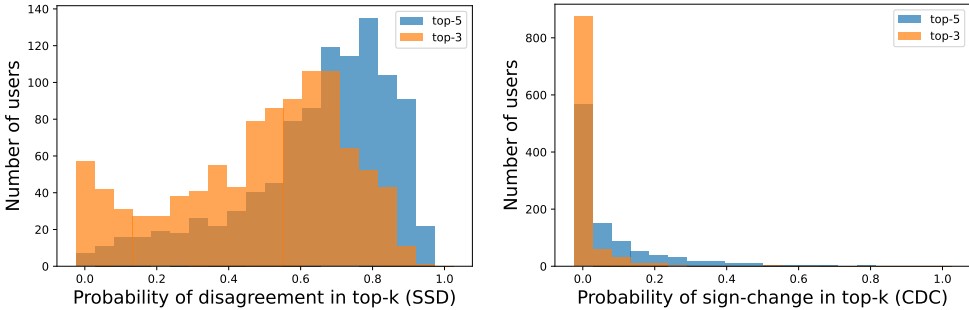

Figure 4: Consistency of explanations. We compare the top-k Shapley values across approximately equivalent models for 1000 users in the hold-out set. [Left] Histogram over the probability of inter-model explanation disagreement (SSD) for each user. [Right] Histogram over the probability of inter-model contradiction (CDC) in the top-k Shapley values for each user. [Shapley type: $\mathcal{D}_{\text{input}}$; dataset: UCI Credit Card; source of indeterminacy: underspecification]

**Consistency Metrics**     The signed set disagreement (SSD) and the contradicting direction of contribution (CDC) metrics are plotted in a (per-user) histogram in Figure 4. They are also averaged over 1000 users and reported in Table 1. In general, we see considerable inconsistency. If a user were shown explanations of a prediction they received from two approximately equivalent models, the probability that the set of top-3 Shapley disagree is over 35% in all our experimental settings, and over 50% in two thirds of them. The odds of a contradiction vary considerably across the experimental settings. They are remarkably high in some of the Rashomon sets, but roughly an order of magnitude lower in the underspecification sets. Nonetheless, the odds of a contradiction in the top-5 Shapley values is over 10% in three quarters of our experimental settings, and over 20% in half of them, potentially alarming levels in a real-world setting.

**Visualizing (in)consistency**     Explanations are often reported in a visual form. We explore the visual consistency of the explanations across approximately equivalent models. To do so, we sort users by the sign agreement (SA) in their top-5 Shapley values across pairs of approximately equivalent models. High SA means the model explanations tend to agree. We select a user, then we plot the explanation for each model within an approximately equivalent set as a separate trace. We limit the

Table 1: Consistency of explanations

| Shapley type: $\mathcal{D}_{\mathrm{mean}}$ | UCI Credit Card | | Give Me Credit | | P.S.'s Safe Driver | |
| --- | --- | --- | --- | --- | --- | --- |
| | top-3 | top-5 | top-3 | top-5 | top-3 | top-5 |
| **Rashomon effect** | | | | | | |
| *Prob. of disagreement (SSD) (%)* | 83.5 | 92.8 | 66.9 | 70.1 | 78.4 | 92.1 |
| *Prob. of contradiction (CDC) (%)* | 46.6 | 62.7 | 6.6 | 19.5 | 19.2 | 34.2 |
| **Underspecification** | | | | | | |
| *Prob. of disagreement (SSD) (%)* | 72.1 | 88.2 | 54.1 | 63.3 | 48.1 | 67.7 |
| *Prob. of contradiction (CDC) (%)* | 7.9 | 20.1 | 3.4 | 13.5 | 0.8 | 2.5 |
| Shapley type: $\mathcal{D}_{\mathrm{input}}$ | | | | | | |
| **Rashomon effect** | | | | | | |
| *Prob. of disagreement (SSD) (%)* | 79.4 | 90.3 | 59.9 | 69.8 | 76.6 | 89.2 |
| *Prob. of contradiction (CDC) (%)* | 36.5 | 54.9 | 11.1 | 29.0 | 13.0 | 23.6 |
| **Underspecification** | | | | | | |
| *Prob. of disagreement (SSD) (%)* | 51.1 | 67.2 | 35.6 | 47.1 | 43.4 | 60.9 |
| *Prob. of contradiction (CDC) (%)* | 2.3 | 9.3 | 2.4 | 10.4 | 0.2 | 0.6 |

plot to the features that are in the top-5 for at least one model. An example of such a plot for the user in the 10th percentile of SA can be seen in Figure 5 (left). We see a contradiction in the explanations for feature "pay-2". For some models, it would be explained to the user that this feature was among the top-5 reasons for a decision, but if an approximately equivalent model were used instead, it could be explained that the same feature was having the *opposite* effect on their decision.

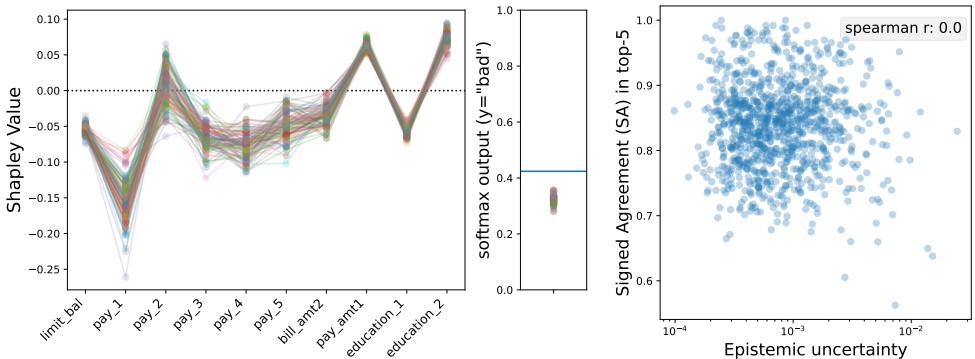

Figure 5: [Left and Center] Visualizing inconsistencies in explanations. Each trace corresponds to one model's explanation: The Shapley values for a subset of the most important features. Notice that "pay_2" crosses the 0 line. For some models it would be presented among the top-5 reasons that a decision was made, yet for other approximately equivalent models, it would be explained as having the *opposite* effect on that decision. In the center, we show the models' softmax outputs. The models tend to agree in their predictions, but not in their explanations. [Right] Predictive vs. explanatory multiplicity. Here we illustrate the relationship between predictive multiplicity and explanatory multiplicity. For 1000 users in the holdout set, we plot the mean intersection of top-5 Shapley values (between pairs of model instances) vs. the (log) epistemic uncertainty in the predictions. The lack of correlation illustrates how epistemic uncertainty is not predictive of consistency in explanation. [Shapley type: $\mathcal{D}_{\mathrm{input}}$; Dataset: UCI Credit Card; Source of indeterminacy: underspecification]

**Variation in predictions vs. variation in explanations**     Within our set of approximately equivalent models, we explore the relationship between the variation in the predictions for a user and the variation in their explanations. We do not observe a consistent relationship across our experimental setups. There are users with low explanatory agreement and high predictive agreement, and vice versa. To illustrate this, we plot the sign agreement (SA) in the top-5 Shapley values vs. the epistemic uncertainty in the associated predictions (computed via Equation 3) for 1000 users. See Figure 5

(right). We also compute the Spearman correlation coefficient ($\rho$). In the UCI Credit Card and Safe Driver datasets, there is effectively no correlation; $\rho$ ranges from -0.17 to 0.23. In the Give Me Some Credit dataset there is weak to moderate negative correlation; $\rho$ ranges from -0.34 to -0.58. One may wish to characterize the uncertainty in an explanation stemming from model indeterminacy. However, our results suggest that the epistemic uncertainty in a prediction is not a reliable indicator of the uncertainty in an explanation.

**Summary and Analysis**    Our results indicate that **1)** model indeterminacy can lead to substantial inconsistencies in the local explanations of approximately equivalent models, even when models only differ because of underspecification; **2)** variation in an explanation for a user across approximately equivalent models can occur with or without variation in the prediction, i.e., whether the epistemic uncertainty in that user's prediction is high or low.

On the one hand, explanatory multiplicity may be viewed as a consequence of high fidelity explanations, which ought to show a sensitivity to the fluctuations of the predictive functions. However, when we consider providing these explanations to a user, whether as a justification for a decision, or as information they can act on, the implications are concerning. Without some kind of guarantee about how long a particular model instance will remain in use, the explanations may be of very little value, or even be detrimental to the users. (Additional results in Appendix D.)

# 6   Discussion

There are notable societal implications to this work. Consider being denied a loan then explained that your number of existing lines of credit was very detrimental to your application, only to learn that—under a trivially different model—your existing lines of credit would have been explained as having helped your application. How would this affect your trust in the decision-making system? Explanation techniques that use Shapley values have been rigorously motivated, and may represent a substantial operational cost in practice, requiring orders of magnitude more compute than just making a prediction. It is sensible to want them to be robust. We should at least communicate the limits of their robustness to the user, so they understand exactly what is being explained, i.e., we should be explicit about the *scope* of the explanation. We elaborate on this discussion in Appendix A.

There is much to explore in future work. Our theoretical analysis is currently limited to a simple linear class of models; our experiments are limited to just two forms of Shapley value explanations. It would be important to explore the empirical effects of model indeterminacy on other explanation methods including recourse. Notably, there is a wide range in the magnitude of the explanatory multiplicity measured across our current experimental settings. It would be valuable to understand the drivers of this variation so it may be preempted in practice. It may also be helpful to put the effects of model indeterminacy into perspective by comparing them to human decision-makers. Human decisions, and the explanations of those decisions, are also liable to fluctuate arbitrarily. However, our results remind us that just because a decision or explanation was produced by a "data driven" model, does not mean that they are fully determined by the data. We hope our work inspires a focused research effort into developing explanation techniques that are more robust to the trivial or arbitrary model choices that occur in practice.

## Acknowledgments and Disclosure of Funding

Resources used in preparing this research were provided, in part, by the Province of Ontario, the Government of Canada through NSERC and CIFAR, and companies sponsoring the Vector Institute[5]. We would like to thank the Vector Institute's computing and administrative staff for their support, the anonymous reviewers for their insightful feedback, and Elliot Creager and Isaac Waller for their helpful comments.

---

[5]https://vectorinstitute.ai

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
