# OpenReview forum: "Implications of Model Indeterminacy for Explanations of Automated Decisions"
_NeurIPS.cc/2022/Conference — NeurIPS 2022 Accept_

### Official Review · Reviewer_77bi · 2022-07-07

**Rating:** 7
**Confidence:** 4
**Soundness:** 3 good
**Presentation:** 4 excellent
**Contribution:** 3 good

**Summary:**

The paper introduces a notion of explanation multiplicity. This is similar to previous works that look at the effects of model indeterminacy such as predictive multiplicity. The authors first show how explanations can vary for models with similar performance on toy examples and theoretically in the linear case. The authors demonstrate that model multiplicity can have noticeable effects on the explanations using shapley sampling explanations, and this can lead to the shapley explanations contradicting in practice for models IRL.

**Questions:**

* how many samples do you use to compute explanations?
* Sampling-based approaches introduce instability themselves and do you consider whether the contradictions come from not sampling enough or explanation multiplicity?

**Limitations:**

Authors have done a sufficient job

**Strengths And Weaknesses:**

Strengths:
- This paper is very well written and easy to follow. It was a pleasure to read, and I had no issues understanding the work.
- The intuition the paper builds with toy examples before providing real experiments is nice. I enjoyed this aspect.
- The idea is well-founded. Previous works on things like predictive multiplicity are quite concerning and this work fits into that literature nicely.
- The fig. 1 is quite nice, very clear + descriptive!
- This work also makes sense in the context of explanations, where aspects like stability have been studied extensively in the past (i.e., small perturbations leading to noticeable differences or simply rerunning explanations leading to large differences). So, this work is a nice extension in that regard to a related, but different notion about how explanations that seemingly should be the same are in fact different.
- The experiments are clear

Weaknesses:

My main critique of this work is that the effects of multiplicity, in this case contradicting explanations, is quite a practical concern, and while the authors have shown this can appear in some cases, the extent to which explanation multiplicity appears in practice is still somewhat left unanswered.  The authors empirically demonstrate their technique using a sampling shap-based approach with the mean as the baseline distribution. Choices around the baseline distribution vary pretty widely and often other variants of shap are used like treeshap + deepshap practically because they're more efficient. It's nice that this work provides evidence this can be a problem in the case presented, but it would be really compelling to evaluate the degree to which explanation multiplicity occurs in other variants of shap that are practically used and other choices of baselines because I suspect readers will be looking for their IRL use cases.

This paper is a weak accept for me because I think the work is quite solid, but extending the evaluation to additional explanation use cases would make the impact much higher.

---

> ### Author Response · Authors · 2022-08-02
> **Response to Reviewer 77bi**
>
> We thank the reviewer for their comments and feedback. We address some of their concerns below:
>
> > My main critique of this work is that the effects of multiplicity, in this case contradicting explanations, is quite a practical concern, and while the authors have shown this can appear in some cases, the extent to which explanation multiplicity appears in practice is still somewhat left unanswered.
>
> We agree that the phenomena the paper explores are most interesting and salient in a practical setting. The catalyst for the work was actually encountering explanatory multiplicity in a practical setting, and being confused about what we were seeing. We do intend to continue this line of investigation to better understand the extent to which explanatory multiplicity appears in other model types and explainability methods. We also aim to better understand the circumstances which lead it to be more/less severe. However, as it stands the paper represents a considerable research effort, and we thought the core ideas merited dissemination. We also wanted to get some feedback from the community before undertaking more complex experiments (additional datasets, models, explainability techniques), and before approaching partners for an industry case study.
>
> > The authors empirically demonstrate their technique using a sampling shap-based approach with the mean as the baseline distribution. Choices around the baseline distribution vary pretty widely and often other variants of shap are used like treeshap + deepshap practically because they're more efficient. It's nice that this work provides evidence this can be a problem in the case presented, but it would be really compelling to evaluate the degree to which explanation multiplicity occurs in other variants of shap that are practically used and other choices of baselines because I suspect readers will be looking for their IRL use cases.
>
> We agree that including other variants of SHAP would make the paper stronger. As mentioned in our response to Reviewer RULt, we are working on this, and in the next version of the paper, we plan to include some additional experiments on variations of the Shapley method we used. For instance, we are beginning to run experiments with 100-reference point baselines, instead of just the mean baseline. (But they take 100 times longer, and therefore they may be limited to a subset of the datasets and sets of approximately equivalent models.)
>
> > How many samples do you use to compute explanations? Sampling-based approaches introduce instability themselves and do you consider whether the contradictions come from not sampling enough or explanation multiplicity?
>
> Since we use a mean baseline in our experiments, the sampling is only used to reduce the number of terms in the combinatorial sum in Equation 1. We use a simple antithetic method, and we use 100 sample-pairs (thereby capturing 200 terms from the sum). We found that this was enough for the relative order of the Shapley values to stabilize. Importantly, we were very careful to ensure that for a given point/user, the explanations across all the models were computed with the exact same set of samples (i.e. the same terms from the sum). While sampling may cause the explanations to deviate slightly from what they would be if we included every term in the sum, all the models are explained using the same set of samples, so it is not what is driving the inter-model explanatory disagreement. We are very sure about this because we also ran experiments on lower dimensional datasets (with less than 12 features) where every term from the sum in Equation 1 can be included. Explanatory multiplicity was very much present in this setting as well. We can include some of these experiments in the Appendix.

---

> > ### Comment · Reviewer_77bi · 2022-08-03
> > **Thanks for the response**
> >
> > Thanks for the author's response, and I appreciate the considerable effort in the research so far. Looking again and considering the response, I think this paper is in quite a good shape, so I raised my score to a seven.

---

### Official Review · Reviewer_RULt · 2022-07-12

**Rating:** 6
**Confidence:** 3
**Soundness:** 3 good
**Presentation:** 3 good
**Contribution:** 3 good

**Summary:**

This paper studies how model indeterminacy impacts explanation multiplicity. Two aspects of model indeterminacy are considered:  predictive multiplicity (underspecification, rashomon sets), and epistemic uncertainty. As models change, explanations are bound to change as the authors show via experiments. However, authors also show that predictive multiplicity or epistemic uncertainty are not good indicators of the extent to which the explanations agree.

There has been prior work on disagreement between explanations by Lakkaraju et al. where different local explainability methods with respect to a single ML model. There is also follow up bayesian work on quantifying the uncertainty of local explanations by the same authors. Rashomon sets have been used by Rudin et al as the authors point.

**Questions:**

a) Can you clarify the key results ?

b) Can you clarify the 3 terms better: predictive multiplicity, model indeterminacy, and epistemic uncertainty ?

c) How do Rashomon sets as computed by you differ from the ones defined in Rudin, et al?

c) How is epistemic uncertainty computed here. Are Rashomon sets or underspecification sets used to compute it ?

d) How would this work/results generalise to other explanation methods -- LIME, counterfactual techniques, etc.

e) Are there any results for case when predictions of multiple almost equivalent models disagree for a user, how do the explanations disagree in this setting.


**Ethics Review Area:**

["I don’t know"]

**Limitations:**

See above.

**Strengths And Weaknesses:**

The problem studied by authors is very useful one, especially in practise. The experiments are quite thorough and good. However the results and their implications are perhaps less clear.

When multiple almost equivalent models have the same prediction for a user, even then local explanations/gradients tend to be different. This is a useful insight. Authors hypothesise that this may be due to the fact that explanations depend on the shape of the predictive function, also seen visually for experiments on small problem sizes. The almost equivalent models are characterised using Rashomon sets and underspecification sets (randomness in training). The results are for one local explanation method -- shap explainer that approximates shapley values.

The related insight is a bit confusing: i.e. the extent to which the explanations disagree is not determined by the extent to which the predictions of multiple almost equivalent models differ. Is this correct ? Pls clarify

The other insight is: If we consider epistemic uncertainty, this might not be a good predictor of explanation multiplicity for that user across the almost equivalent models. Is this correct ? Pls clarify

The main issues with the paper are:
(a) Lack of clarity on following 3 terms: predictive multiplicity, model indeterminacy, and epistemic uncertainty.
(b) Lack of clarity on results.
(c) Experimental results are based on only 1 local explanation method:  SHAP.
(d) The analytical results are based on linear setting alone.

---

> ### Author Response · Authors · 2022-08-02
> **Response to Reviewer RULt (part 2)**
>
> > The analytical results are based on linear setting alone.
>
> We agree this would be more interesting in a more general case. We will likely try to explore that in future work.
>
> > How do Rashomon sets as computed by you differ from the ones defined in Rudin, et al?
>
> Semenova, Rudin, and Parr, define the empirical Rashomon set (or simply Rashomon set) as the set of all models in the hypothesis space having empirical risk within some small epsilon > 0 of the optimal risk (achieved by the empirical risk minimizer). We use this definition in our linear analysis.
>
> What we call “Rashomon effect sets” in our experiments differ in two ways: 1) We used the AUROC on a validation set to determine which models were in the Rashomon set, not the empirical risk. We felt this was closer to what would be used in a practical model selection process. 2) We consider only a finite sample of models, those which were found during a realistic model sweep. We do not consider all models in the hypothesis space as it is impractical in our settings (neural nets or other complex hypothesis spaces). We can describe these differences in the Appendix, and point the reader to them at the top of page 7.
>
> > How is epistemic uncertainty computed here? Are Rashomon sets or underspecification sets used to compute it ?
>
> Epistemic uncertainty is computed on both the Rashomon sets and the underspecification sets. Whenever we discuss/report the epistemic uncertainty for a set of approximately equivalent models, it was computed over the models in that set, and using Equation 2. We can clarify this. We can also add some sample calculations in the Appendix, since it is a little unclear how Equation 2 should be applied to a set of models.
>
> If we want the epistemic uncertainty at input $x$ we solve Equation 2 for the epistemic term.
> Let $f_n(x)$ be the output probability of $y=1$ (a "bad" outcome) from model $n$ for an input/user $x$.
> Let $H(f_n(x))$ be the entropy of that output probability. Then the epistemic uncertainty at $x$ over a set of $N$ models is computed as: $H(\frac{1}{N}\sum_n f_n(x)) - \frac{1}{N}\sum_n H(f_n(x))$. The entropy of the mean minus the mean of the entropies. A set of four models having output probabilities [0.52, 0.51, 0.49, 0.48] at $x$ would have low epistemic uncertainty. While a set having outputs [0.99, 0.98, 0.02, 0.01] at $x$ would have high epistemic uncertainty.
>
> > Are there any results for cases when predictions of multiple almost equivalent models disagree for a user, how do the explanations disagree in this setting ?
>
> Our results are not limited to points/users where the model predictions agree. So this question is probably best answered with Figure 5 (right) and Figure 10 in the Appendix. The x-axis shows epistemic uncertainty (which we use to quantify predictive disagreement), the y-axis shows a measure of explanatory agreement (which we will clarify–see our response to Reviewer UP6W).
> If we only consider the points/users on the right, where epistemic uncertainty is high, and hence the models disagree, we can get a sense of the distribution of explanatory agreement. We can then contrast this to the points/users on the left, where epistemic uncertainty is low. By doing so we see that epistemic uncertainty (predictive disagreement) is not indicative of explanatory agreement. There is some very weak correlation, but it is not consistent. We will clarify this in the Results section.

---

> ### Author Response · Authors · 2022-08-02
> **Response to Reviewer RULt (part 1)**
>
> We thank the reviewer for their comments and feedback. We address some of their concerns below:
>
> > The experiments are quite thorough and good. However the results and their implications are perhaps less clear. Can you clarify the key results ?
>
> We agree the key results could be clarified and better summarized. We wil adjust the Results section to do so. In our opinion the key results are: 1) Model indeterminacy can lead to substantial inconsistencies in the local explanations of approximately equivalent models, even when models only differ because of underspecification. 2) Point-wise measures of inter-model prediction agreement, like the computation of epistemic uncertainty over a set of models, do not indicate where the local explanations will agree/disagree. 3) This could happen in a setting of material consequence (credit, insurance), where it could have adverse consequences on a user, e.g., misleading explanations, confusion, frustration.
>
> > The related insight is a bit confusing: i.e. the extent to which the explanations disagree is not determined by the extent to which the predictions of multiple almost equivalent models differ. Is this correct ? Pls clarify
>
> Yes, that is correct. Our experiments show little to no correlation between inter-model prediction agreement at a point, and the inter-model agreement in the explanations at that same point.
>
> > If we consider epistemic uncertainty, this might not be a good predictor of explanation multiplicity for that user across the almost equivalent models. Is this correct ? Pls clarify
>
> Yes, that is correct. We use epistemic uncertainty, computed point/user-wise over the set of models, as our principal measure of inter-model predictive agreement. High epistemic uncertainty is indicative of low inter-model agreement and vice-versa. The reason we don’t measure predictive agreement by simply thresholding the model output probabilities is because that approach is quite sensitive to the threshold used, and in a risk assessment setting (like credit or insurance) threshold selection is a function of risk appetite. We discuss this in Appendix D.1.
>
> > The main issues with the paper are: Lack of clarity on following 3 terms: predictive multiplicity, model indeterminacy, and epistemic uncertainty. Can you clarify the 3 terms better: predictive multiplicity, model indeterminacy, and epistemic uncertainty ?
>
> Admittedly, we use the term predictive multiplicity quite a lot, and offer little in the way of a definition beyond a reference to the work where the term was defined (Marx et al., 2020). They define predictive multiplicity as “the ability of a prediction problem to admit competing models with conflicting predictions”. We will include their original definition, and tweak our usage of the term throughout the paper to more precisely match it.
>
> Model indeterminacy is a term we introduce to unify the Rashomon effect and underspecification. The Rashomon effect includes underspecification, but we thought an umbrella term would be useful for discussions that contrast the two phenomena. Importantly, we did not want the take-home message to be that the Rashomon effect can lead to contradicting explanations, because in our opinion the more surprising/salient result is that underspecification alone can lead to contradicting explanations. We dedicate Section 2.1 to a description of model indeterminacy, but we will work on clarifying and improving this. We will also simplify Figure 2, and use it to more clearly relate the concepts of model indeterminacy, underspecification, and the Rashomon effect.
>
> Epistemic uncertainty is used principally as a measure of inter-model predictive agreement. (As discussed above). We dedicate most of Section 2.3 to the description of epistemic uncertainty. But we can do a better job of tying it back to predictive multiplicity and the inter-model predictive agreement we aim to quantify with it. We will also include sample calculations in the Appendix.
>
> > Experimental results are based on only 1 local explanation method: SHAP. How would this work/results generalize to other explanation methods -- LIME, counterfactual techniques, etc.?
>
> We agree that this is an interesting question, and we have begun to explore it. Specifically, we have run some experiments to measure how model indeterminacy affects certain algorithmic recourse methods (which are closely related to counterfactual explanations). In short, we are finding that recourse (or a counterfactual) computed on one model can be invalid in an approximately equivalent model. We may mention this finding. However, there are enough subtleties with recourse that we think it deserves separate treatment.
>
> We have not yet explored LIME. However, in the next version we plan to include some additional experimental results on variations of the Shapley method we used.

---

### Official Review · Reviewer_UP6W · 2022-07-12

**Rating:** 8
**Confidence:** 3
**Soundness:** 4 excellent
**Presentation:** 4 excellent
**Contribution:** 4 excellent

**Summary:**

The authors show how model multiplicity (i.e. different models which have similar test set performance) can result in explanation multiplicity.  Using Shapely values for feature importance, often models that are underspecified (i.e. have the same hyper parameters setup but different initializations) but have similar test performance can result in different feature importance rankings and even entirely contradictory ones. The authors compare Rashomon effect (vastly different hyper parameters, similar test performance) and underspecification (same hyper parameters, different inits, similar test performance). The Rashomon effect has a much greater impact on agreement and contradiction (as expected), but underspecification also results in fairly high levels of contractiction.

**Questions:**

The authors note "The odds of a contradiction vary considerably across the experimental settings. They are remarkably high in some of the Rashomon sets, but roughly an order of magnitude lower in the underspeciﬁcation sets", could the authors speculate as to why this is. Are the models in the Rashomon set drastically different? Also, why are the contradict percentages fro the UCI Credit Card dataset in the top-3 so much higher than the other datasets? Did the models for that dataset evaluate more poorly on the test datasets?

**Limitations:**

The authors show how the arbitrary selection between similarly performing models can results in drastically different explanations. They couch the work in datasets related credit and insurance, an area which providing good explanations would allow the individual to modify their behavior such that they can score higher. In their work they show how the initial explanation would not often hold-up if a different (but similarly performing) model had been selected.

**Strengths And Weaknesses:**

Very strong experimental design tackling a very important issue. Results felt rushed, things are pushed to the appendix that should have been kept in the main paper. I found Figure 2 to be confusing, but the caption was very clear. Also, the right hand side of Figure 5. Overall, paper is very well written and the work is very significant.

---

> ### Author Response · Authors · 2022-08-02
> **Response to Reviewer UP6W**
>
> We thank the reviewer for their comments and feedback. We address some of their concerns below:
>
> > I found Figure 2 to be confusing, but the caption was very clear.
>
> We agree that Figure 2 is quite busy and should be simplified to better reflect the caption. It was designed to also support a discussion about the scope of an explanation that was moved to Appendix A, but it would be clearer if we used two separate figures. We will change the figure accordingly in the next version of the paper.
>
> > Also, the right hand side of Figure 5.
>
> We realize that in an effort to conserve space, we omitted an explanation of what is depicted on the vertical axis of the scatter plot: “the mean overlap in the top-5 Shapley values”. We will clarify that this is an empirical mean (over pairs of model instances in the set) of the intersection of top-5 Shapley values. Each point in the scatter plot is a point in the test set. The lack of correlation shows that epistemic uncertainty is not predictive of consistency in explanation.
>
> > "The odds of a contradiction vary considerably across the experimental settings. They are remarkably high in some of the Rashomon sets, but roughly an order of magnitude lower in the underspeciﬁcation sets", could the authors speculate as to why this is
>
> We understand this to be because of a difference in the functional diversity in each set type. The model instances in the underspecification sets come from a subset of the hypothesis space searched in the Rashomon effect sets. Therefore it is likely that the Rashomon effect sets include a more diverse set of predictive functions than just those in the underspecification sets.
>
> > Also, why are the contradict percentages fro the UCI Credit Card dataset in the top-3 so much higher than the other datasets? Did the models for that dataset evaluate more poorly on the test datasets?
>
> This is a very good question. The AUROC performance of the models in the various sets are presented in Table 2 in Appendix B.2, and the consistency of the model decisions are reported in Table 3 in Appendix D.1. The models trained on the UCI Credit Card rank in the middle (compared to the two other datasets) in terms of AUROC performance, and highest in terms of decision consistency. (Decisions having the lowest chances of “flipping” when switching between any two approximately equivalent models). So it is not as simple as the models evaluating more poorly, or disagreeing more in their predictions.
>
> The short answer is that we do not yet know what explains this difference, but we are investigating the potential factors. It may be due to a strong statistical dependence between features, a situation along the same lines as the “strong covariance” situation described in the linear analysis, but in the more general nonlinear case. We think it would be very valuable to know when a data scientist or engineer should expect to see contradictory explanations across approximately equivalent models, and we see this as an interesting direction for future work.

---

> > ### Comment · Reviewer_UP6W · 2022-08-10
> > **Changes to figure.**
> >
> > Thank you for addressing my questions. I appreciate you all improving the figures to make it clearer.

---

### Meta-Review · Area_Chair_9WKw · 2022-08-30

**Recommendation:** Accept
**Confidence:** Certain

**Metareview:**

This paper introduces the notion of explanation multiplicity. The authors first show how explanations can vary for models with similar performance on toy examples and theoretically in the linear case. They demonstrate that model multiplicity can have noticeable effects on the explanations using Shapley explanations. This paper puts forth a strong approach tackling a very important issue. So, all the reviewers concur that this paper should be accepted at this time.

**Award:**

No

---

### Decision · Program_Chairs · 2022-09-14

Accept